# A Fair and In-Depth Evaluation
# of Existing End-to-End Entity Linking Systems

**Hannah Bast**[1*] and **Matthias Hertel**[1,2*] and **Natalie Prange**[1*]

[1]University of Freiburg, Department of Computer Science, Germany
[2]Karlsruhe Institute of Technology, Institute for Automation and Applied Informatics, Germany
{bast,prange}@cs.uni-freiburg.de     matthias.hertel@kit.edu

## Abstract

Existing evaluations of entity linking systems often say little about how the system is going to perform for a particular application. There are two fundamental reasons for this. One is that many evaluations only use aggregate measures (like precision, recall, and F1 score), without a detailed error analysis or a closer look at the results. The other is that all of the widely used benchmarks have strong biases and artifacts, in particular: a strong focus on named entities, an unclear or missing specification of what else counts as an entity mention, poor handling of ambiguities, and an over- or underrepresentation of certain kinds of entities.

We provide a more meaningful and fair in-depth evaluation of a variety of existing end-to-end entity linkers. We characterize their strengths and weaknesses and also report on reproducibility aspects. The detailed results of our evaluation can be inspected under https://elevant.cs.uni-freiburg.de/emnlp2023. Our evaluation is based on several widely used benchmarks, which exhibit the problems mentioned above to various degrees, as well as on two new benchmarks, which address the problems mentioned above. The new benchmarks can be found under https://github.com/ad-freiburg/fair-entity-linking-benchmarks.

## 1 Introduction

Entity linking is a problem of fundamental importance in all kinds of applications dealing with natural language. The input is a text in natural language and a knowledge base of entities, each with a unique identifier, such as Wikipedia or Wikidata. The task is to identify all sub-sequences in the text that refer to an entity, we call these *entity mentions*, and for each identified entity mention determine the entity from the knowledge base to which it refers.

Here is an example sentence, with the entity mentions underlined and the corresponding Wikidata ID in square brackets (and clickable in the PDF):

American [Q30] athlete Whittington [Q21066526] failed to appear in the 2013–14 season [Q16192072] due to a torn ACL [Q18912826].

For research purposes, the problem is often split in two parts: *entity recognition* (ER; identifying the entity mentions) and *entity disambiguation* (ED; identifying the correct entity for a mention). In practical applications, the two problems almost always occur together. In this paper, we consider the combined problem, calling it *entity linking* (EL)[1].

### 1.1 Problems with existing evaluations

There is a huge body of research on entity linking and many systems exist. They usually come with an experimental evaluation and a comparison to other systems. However, these evaluations often say little about how the system will perform in practice, for a particular application. We see the following two fundamental reasons for this.

**Coarse evaluation metrics.** Most existing evaluations compare systems with respect to their precision, recall, and F1 score; we call these *aggregate measures* in the following. In particular, the popular and widely used GERBIL platform (Röder et al., 2018) supports only comparisons with respect to (variants of) these measures.[2] What is often missing is a detailed error analysis that compares the linkers along meaningful error categories. This often results in linkers that perform well on the selected benchmarks (critically discussed in the next paragraphs), but not in other applications. On top of that, we also had considerable problems with just replicating the reported results.

---

*Author contributions are stated in Section 8. M.H. is funded by the Helmholtz Association's Initiative and Networking Fund through Helmholtz AI.

[1]We deliberately do not refer to these problems as NER, NED and NEL. We omit the "N(amed)" because an important aspect of our evaluation is that we consider non-named entities as well.

[2]GERBIL does support separate evaluation of ER and ED, but again with respect to these aggregate measures only.

**Benchmark artifacts and biases.** The following four artifacts and biases are frequent in existing benchmarks. Linkers can exploit these to achieve good results, especially regarding the aggregate measures discussed in the previous paragraph.

First, all widely used benchmarks have a *strong focus on named entities*, which in the English language are almost always capitalized and hence easy to recognize. However, many if not most entity-linking applications need to recognize more than just named entities, for example: professions ("athlete"), chemical elements ("gold"), diseases ("torn ligament"), genres ("burlesque"), etc.

Second, when going beyond named entities, it is hard to *define what counts as an entity mention*. Existing benchmarks work around this problem in one of three ways: they contain almost exclusively named entities, the decision was up to annotators without clear guidelines and without documentation, or it is expected from the evaluation that the entity mentions are fixed and only the disambiguation is analyzed. Note that it is not an option to call anything an entity that has an entry in a knowledge base like Wikipedia or Wikidata, because then almost every word would become part of an entity mention.[3]

Many *entity mentions are ambiguous*, making it debatable which entity they should be linked to. A typical example is the mention *American* in the sentence above. There is no Wikipedia or Wikidata entry for the property of being American. Instead, there are three closely related entities: the country [Q30], the language [Q7976], and the citizens [Q846570]. Most existing benchmarks resort to one choice, which punishes systems that make an alternative (but maybe equally meaningful) choice.

Several benchmarks have a strong *bias towards certain kinds of entities*. A prominent example is the widely used *AIDA-CoNLL* benchmark (Hoffart et al., 2011). It contains many sports articles with many entities of the form *France*, where the correct entity is the respective sports team and not the country. This invites overfitting. In particular, learning-based systems are quick to pick up such signals, and even simple baselines can be tuned relatively easily to perform well on such benchmarks.

We are not the first to recognize these problems or try to address them. In fact, there have been several papers in recent years on the meta-topic of a more meaningful evaluation of entity linking systems. We provide a succinct overview of this work in Section 2. However, we have not found any work that has tried to address *all* of the problems mentioned above. This is what we set out to do in this paper, by providing an in-depth comparison and evaluation of the currently best available entity linking systems on existing benchmarks as well as on two new benchmarks that address the problems mentioned above.

## 1.2 Contributions

We provide an in-depth evaluation of a variety of existing end-to-end entity linkers, on existing benchmarks as well as on two new benchmarks that we propose in this paper, in order to address the problems pointed out in Section 1.1. More specifically:

• We provide a detailed error analysis of these linkers and characterize their strengths and weaknesses and how well the results from the respective publications can be reproduced. See Table 1 and Figure 1 for an overview of our results, Table 4 and Section 6 for the details, and Section 7 for a concluding summary of the main takeaways. Detailed individual results of our evaluation can be inspected under https://elevant.cs.uni-freiburg.de/emnlp2023.

• We describe the most widely used existing benchmarks and reveal several artifacts and biases that invite overfitting; see Section 4. We create two new benchmarks that address these problems; see Section 5. These benchmarks can be found under https://github.com/ad-freiburg/fair-entity-linking-benchmarks.

## 2 Related Work

Ling et al. (2015) analyze differences between versions of the entity linking problem that are being tackled by different state-of-the-art systems. They compare popular entity linking benchmarks and briefly discuss inconsistent annotation guidelines. However, they do not present improved benchmarks. They develop a modular system to analyze how different aspects of an entity linking system affect performance. They manually organize linking errors made by this system into six classes to gain a better understanding of where linking errors occur. We use the more fine-grained error categories introduced by Bast et al. (2022) for a thorough comparison between linking systems.

---

[3]For example, there is a Wikipedia article for the grammatical article *the* or the general concepts of *failure* or *appearance*, all used in our example sentence.

| System | Overall F1 | ER F1 | Disamb. accuracy | Strengths and Weaknesses | Repro-ducibility |
|---|---|---|---|---|---|
| ReFinED | 73.3% | 82.7% | 89.2% | very good overall results; particularly strong on metonyms | good |
| REL | 67.7% | 82.3% | 83.0% | very high ER F1; often falsely links NIL mentions | very good |
| GENRE | 64.6% | 74.2% | 87.4% | sacrifices ER recall for high disambiguation accuracy | mediocre |
| Ambiverse | 59.0% | 76.2% | 78.3% | good on partial names; detected spans often too short | problematic |
| Neural EL | 50.6% | 73.6% | 68.7% | good on demonyms; struggles with partial names | mediocre |
| Baseline | 46.3% | 74.0% | 63.8% | predicts entity with highest prior probability; ignores context | - |
| TagMe | 43.0% | 54.2% | 80.7% | high disambiguation accuracy; poor ER | poor |

Table 1: Overview of the results of the evaluation. Scores are given as unweighted average over all five benchmarks (that is, the score for each benchmark contributes equally to the average, and is independent of the number of mentions in that benchmark).

Rosales-Méndez et al. (2019) also aim for a fairer comparison between entity linking systems. They create a questionnaire to examine the degree of consensus about certain annotation decisions in the EL community. Based on the results of their questionnaire they create a fine-grained annotation scheme and re-annotate three existing benchmarks accordingly. They add new annotations to capture as many potential links as possible. Additionally, they annotate some mentions with multiple alternatives. They define two annotation modes, strict and relaxed, where the former includes only named entities and the latter includes all entities that can be linked to Wikipedia. Their approach is more extreme than ours in several respects: their relaxed mode contains very many annotations, (because of that) they consider only smaller benchmarks, and their error categories are very fine-grained. Furthermore, they evaluate only older linkers.

Jha et al. (2017) identify inconsistencies between EL benchmarks and define a set of common annotation rules. They derive a taxonomy of common annotation errors and propose a semi-automatic tool for identifying these errors in existing benchmarks. They then create improved versions of current benchmarks and evaluate the effects of their improvements with 10 different ER and EL systems. However, their annotation rules are made without properly addressing the disagreement about them in the entity linking community. For our benchmark generation, we instead opt to allow multiple alternative annotations in cases where a good argument can be made for any of these linking decisions.

Van Erp et al. (2016) analyze six current entity linking benchmarks and derive suggestions for how to create better benchmarks. They examine different benchmark aspects: (1) the document type (2) entity, surface form and mention characteristics and (3) mention annotation characteristics. They suggest to document decisions that are being made while creating the benchmark, which includes annotation guidelines. Apart from that, they do not provide guidelines or suggestions that target the annotation process.

Brasoveanu et al. (2018) argue that an in-depth qualitative analysis of entity linking errors is necessary in order to efficiently improve entity linking systems. They categorize EL errors into five categories: knowledge base errors, dataset errors, annotator errors, NIL clustering errors and evaluation errors. They select four systems and three benchmarks and manually classify errors into these categories. Their evaluation is very short, and their main result is that most errors are annotator errors.

Ortmann (2022) raises the issue of double penalties for labeling or boundary errors when computing recall, precision and F1 score in the general context of evaluating labeled spans. Namely, an incorrect label or an incorrect span boundary counts as both a false positive and a false negative while, e.g., a prediction that does not overlap with any ground truth annotation counts as only one false positive even though it is arguably more wrong. Ortmann introduces a new way of computing precision, recall and F1 score where such errors do not count double. We use the standard precision, recall and F1 score for our evaluation, but complemented by fine-grained error categories that show the effect of such errors on the overall score.[4]

---

[4]This is in line with our philosophy that a single overall score should be taken with a grain of salt anyway and that one needs to look at the results more closely to determine strengths and weaknesses of a system.

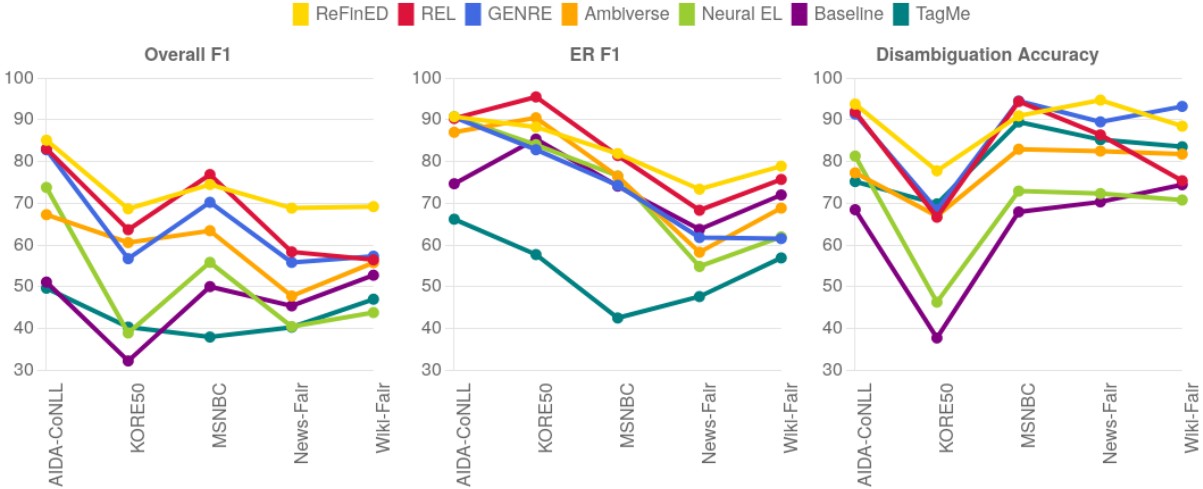

Figure 1: Overall results of each system on each benchmark; see Table 4 for more fine-grained results.

## 3 Metrics

We report micro precision, recall and F1 scores, both for the overall EL task and for the ER subtask. Details for how these measures are computed are provided in Section A.1. Additionally, we use the fine-grained error metrics provided by the evaluation tool ELEVANT (Bast et al., 2022) to analyze the strengths and weaknesses of the evaluated linkers in detail:

**ER false negatives** The following metrics analyze special cases of ER false negatives. *Lowercased*: the number of lowercased mentions that are not detected. *Partially included*: the number of mentions where only a part of the mention is linked to some entity.

**ER false positives** ER false positives are predicted mentions that do not correspond to a ground truth mention or that correspond to a ground truth mention annotated with *NIL*. The following metrics analyze special cases of ER false positives. *Lowercased*: the number of falsely predicted mentions written in lower case. *Ground truth NIL*: the number of predicted mentions that correspond to a ground truth mention annotated with *NIL*. *Wrong span*: the number of predicted mentions that are part of or overlap with a ground truth mention of the predicted entity, but the predicted span is not correct.

**Disambiguation** The disambiguation accuracy is defined as the correctly linked entities divided by the correctly detected entity mentions. We compute fine-grained disambiguation accuracies on several mention categories that are difficult to disambiguate, by only considering ground truth mentions with specific properties. The following categories are analyzed. *Demonym*: the mention appears in a list of demonyms (e.g., *German*).[5] *Metonymy*: the most popular candidate is a location but the ground truth entity is not a location. *Partial name*: the mention is a part of the ground truth entity's name but not the full name. *Rare*: the most popular candidate for the mention is not the ground truth entity. Statistics of the frequencies of these categories across the benchmarks are given in Table 3. We also report the disambiguation error rate, which is simply one minus the disambiguation accuracy.

## 4 Critical review of existing benchmarks

We analyze the performance of the entity linking systems included in our evaluation on three of the most widely used existing benchmarks[6]. It turns out that each of them has its own quirks and biases, as discussed in the following sections. Statistics on the annotated entity mentions for each benchmark are provided in Table 3. See Section A.2 for other popular EL benchmarks that we have excluded from our evaluation due to problems in their design.

### 4.1 AIDA-CoNLL

The AIDA-CoNLL dataset (Hoffart et al., 2011) is based on the CoNLL-2003 dataset for entity

---

[5]Our definition of *demonyms* includes cases where the ground truth mention is a language or ethnicity rather than a group of people, as long as the mention word appears in the list of demonyms.

[6]Note: we map all entities to Wikidata for our evaluation.

recognition which consists of news articles from the 1990s. Hoffart et al. manually annotated the existing proper-noun mentions with corresponding entities in the YAGO2 knowledge base. The dataset is split into train, development and test set. For our evaluation, we use the test set which consists of 231 articles. The benchmark has a strong bias towards sports articles ($44\%$ of articles are sports related). This results in a large amount of demonym and metonym mentions. The average results achieved by the evaluated systems on AIDA-CoNLL are much higher than the average results on all other benchmarks included in our evaluation. Entity mentions in AIDA-CoNLL are mostly easy-to-detect single or two-word mentions (like names). Only $5.5\%$ of mentions consist of more than two words which makes the ER part particularly easy on this benchmark.

## 4.2 KORE50

The KORE50 benchmark (Hoffart et al., 2012) consists of 50 hand-crafted sentences from five domains (celebrities, music, business, sports, politics). The sentences were designed to make entity disambiguation particularly challenging, mainly by using only partial names when referring to persons. Thus, the benchmark contains a lot of partial names and entities of type person. This also entails that, like AIDA-CoNLL, KORE50 contains hardly any mentions with more than two words. In fact, $91.7\%$ of mentions are easy-to-detect single-word mentions.

## 4.3 MSNBC

The MSNBC benchmark (Cucerzan, 2007) consists of 20 news articles from 2007. In our evaluation, we use an updated version by Guo and Barbosa (2018) (the results are usually similar to those on the original benchmark). Cucerzan took the top two stories of the ten MSNBC News categories, used them as input to his entity linking system and then manually corrected the resulting annotations. Adjectival forms of locations are rarely and inconsistently annotated in the benchmark[7]. The original dataset contains overlapping annotations for no obvious reason[8]. This was fixed in the updated version by Guo and Barbosa. They also removed links to no longer existing Wikipedia articles. Sev-

| GT mention property | Wiki-Fair | News-Fair |
|---|---|---|
| All | 1482 | 359 |
| Linked to NIL | 132 | 49 |
| Optional | 447 | 84 |
| Has alternative annotation(s) | 118 | 22 |

Table 2: Number of ground truth mentions with the given properties for our two benchmarks (without coreferences).

eral articles differ from the ones in the original benchmark, but revolve around the same topic.

## 5 Our new fair benchmarks

We create two benchmarks to address the shortcomings observed in existing entity linking benchmarks. The benchmarks are publicly available through our GitHub repository[9]. The first benchmark, Wiki-Fair, consists of 80 randomly selected Wikipedia articles, the second one, News-Fair, of 40 randomly selected news articles from a webnews crawl (Akhbardeh et al., 2021). In each of these articles, three random consecutive paragraphs were manually annotated with Wikidata entities. The rest of the article remains unannotated. This way, a large variety of topics is covered with an acceptable amount of annotation work while still allowing linkers to use complete articles as context. Annotating the benchmarks with Wikidata entities instead of Wikipedia (or DBpedia) entities decreases the likelihood of punishing a linker for correctly linking an entity that was not contained in the knowledge base during benchmark creation, since the number of entities in Wikidata is an order of magnitude larger than in Wikipedia.

We also annotate non-named entities in our benchmarks. In the few existing benchmarks that contain non-named entities, there is typically no discernible rule for which non-named entities were annotated such that the annotations seem rather arbitrary. To address this issue, we define a type whitelist (given in Section A.5) and annotate all entities that have an "instance_of"/"subclass_of" path in Wikidata to one of these types[10].

As discussed by Ling et al. (2015), existing entity linking benchmarks differ significantly in which mentions are annotated and with which enti-

---

[7]While "Iraqi", "German" or "Syrian" are not annotated at all, "U.S." in the phrase "U.S. builder" is annotated, but not in the phrase "U.S. helicopter".

[8]E.g., in the phrase "Frank Blake", both, the entire phrase and "Blake" are annotated separately but with the same entity.

[9]https://github.com/ad-freiburg/fair-entity-linking-benchmarks

[10]E.g., the word "athlete" in the example sentence in Section 1 would be annotated in our benchmarks, since the Wikidata entity for athlete (Q2066131) is an instance of the type "occupation" which is one of our whitelist types.

| Benchmark | mentions | lower | multiword | NIL | demonym | metonym | partial | rare | person | location | organization |
|---|---|---|---|---|---|---|---|---|---|---|---|
| AIDA-CoNLL | 5616 | 0% | 37% | 20% | 6% | 9% | 15% | 11% | 18% | 31% | 53% |
| KORE50 | 144 | 0% | 8% | 1% | 0% | 6% | 61% | 11% | 53% | 11% | 28% |
| MSNBC | 739 | 1% | 43% | 12% | 0% | 2% | 33% | 7% | 32% | 24% | 40% |
| News-Fair | 275 | 24% | 36% | 18% | 0% | 4% | 13% | 15% | 21% | 13% | 26% |
| Wiki-Fair | 1035 | 18% | 43% | 13% | 4% | 0% | 14% | 14% | 21% | 32% | 32% |

Table 3: Statistics about types of mentions and entities in the benchmarks. *mentions*: number of (non-optional) ground truth entity mentions. *lower*: lowercased mentions. *multiword*: mentions that consist of multiple words. *NIL*: mentions where the annotation is *Unknown*. *demonym*: demonym mentions. *metonym*: metonym mentions. *partial*: the mention text is a part of the entity's name (but not the full name). *rare*: the most popular candidate for the mention is not the ground truth entity. *person/location/organization*: entities of type person/location/organization. Note that these entity types can sum up to more than 100% because some entities have more than one type.

| System | ER false negatives | | ER false positives | | | Disambiguation error rates | | | |
|---|---|---|---|---|---|---|---|---|---|
| | lower-cased | partially included | lower-cased | gr. truth NIL | wrong span | demonym | metonym | partial name | rare |
| ReFinED | 39.6 | 14.6 | 6.6 | 121.2 | 11.4 | 5.7% | 30.8% | 16.8% | 17.5% |
| REL | 42.4 | 20.8 | 0.6 | 115.4 | 10.0 | 19.0% | 27.1% | 25.3% | 30.9% |
| GENRE | 44.4 | 16.0 | 1.4 | 52.2 | 13.2 | 2.1% | 28.4% | 19.5% | 15.1% |
| Ambiverse | 43.4 | 33.6 | 22.6 | 121.8 | 15.6 | 39.6% | 73.9% | 29.3% | 43.5% |
| Neural EL | 44.4 | 17.6 | 0.0 | 95.6 | 8.0 | 22.5% | 78.1% | 54.7% | 73.2% |
| Baseline | 41.8 | 37.2 | 56.2 | 110.6 | 10.2 | 53.1% | 100.0% | 65.7% | 100.0% |
| TagMe | 27.8 | 21.4 | 462.6 | 70.8 | 39.4 | 51.5% | 63.4% | 23.4% | 60.0% |

Table 4: Average results over all five benchmarks for the fine-grained evaluation measures defined in Section 3. Note that the error rate is just one minus the accuracy. For "demonym" and "metonym" error rates, only those benchmarks were considered that contain at least 2% of demonyms or metonyms, respectively.

ties. With our benchmarks, we want to introduce a basis for fairer comparison of different approaches by giving annotation alternatives in cases where multiple annotations could be considered correct.[11] We found that the averaged F1 scores of all evaluated linkers are 5.2% lower on Wiki-Fair and 3.7% lower on News-Fair when not providing these alternatives and only annotating the longer mentions.

Since there is considerable disagreement about the definition of a named entity, we introduce the concept of optional ground truth annotations, which includes dates and quantities. A prediction that matches an optional ground truth annotation will simply be ignored, i.e., the system will not be punished with a false positive, but the prediction does not count as true positive either.

We also annotate coreference mentions. How-ever, for the evaluation in this work, we use a version without coreference mentions.

The total number of ground truth mentions is shown in Table 2. The details of our annotation guidelines are given in Section A.4.

## 6 Evaluation of existing entity linkers

In the following we analyze six entity linking systems in detail. Our evaluation includes linkers to which code or an API are available and functional such that linking results can easily be produced[12]. Furthermore, we restrict the set of linkers to those that either achieve strong results on popular benchmarks or are popular in the entity linking community. Table 1 gives an overview of the results for all evaluated systems including a simple baseline that uses spaCy (Honnibal et al., 2020) for ER and always predicts the entity with the highest prior probability given only the mention text. The two systems with the weakest results in our evaluation (Neural EL and TagMe) are discussed in detail in the appendix (A.3). The appendix also contains

---

[11]For example, both linking the entire phrase in "Chatham, New Jersey" to the entity for Chatham and linking just "Chatham" to the entity for Chatham (while linking "New Jersey" to the entity for the state New Jersey) are considered correct on our benchmark. If a system predicts the mentions from the latter case, the prediction counts as a single true positive if and only if both mentions were correctly recognized and linked to the correct entities. Otherwise it is counted as a single FN. This is to avoid the need for fractional TP or FN. FPs are counted as usual.

[12]This excludes for example Kolitsas et al. (2018), see these GitHub issues, Ravi et al. (2021) see these GitHub issues or Broscheit (2019).

a discussion of two systems that we did not include in our table due to very weak results and reproducibility issues. The individual results for all evaluated linkers can be examined in full detail in our ELEVANT instance[13].

## 6.1 ReFinED

Ayoola et al. (2022) developed ReFinED, a fast end-to-end entity linker based on Transformers. They train a linear layer over Transformer token embeddings to predict BIO tags for the ER task. Mentions are represented by average pooling the corresponding token embeddings. They use a separate Transformer model to produce entity embeddings from the label and description of an entity. The similarity between mention and entity embeddings is combined with an entity type score and a prior probability to a final score.

ReFinED comes in two variants: A model trained on Wikipedia only and a model fine-tuned on the AIDA-CoNLL dataset. We report results for the fine-tuned version because it outperforms the Wikipedia version on all benchmarks in our evaluation. Moreover, ReFinED can be used with two different entity candidate sets: 6M Wikidata entities that are also contained in Wikipedia or 33M Wikidata entities. We choose the 6M set because it achieves better results on most benchmarks.[14]

**Evaluation summary** Of the systems included in our evaluation, ReFinED has the best overall F1 score and is strong both for ER and for disambiguation. Its closest competitors are GENRE and REL, which are considerably worse regarding ER (GENRE) or disambiguation (REL).

**Recognition** ReFinED has a generally high ER F1 score, but the performance difference to the other systems is particularly large on Wiki-Fair and News-Fair. This can at least partly be attributed to the fact that, in contrast to most other systems, ReFinED sometimes links lowercased mentions, which are only annotated on our benchmarks.

On AIDA-CoNLL, it has the highest numbers of ER FP for mentions where the ground truth entity is NIL. A closer inspection shows that in many of these cases, the system's predictions are actually correct and the ground truth entity was annotated as NIL, probably due to an incomplete knowledge

base at the time of the annotation. The same trend can not be observed on our most up-to-date benchmarks, Wiki-Fair and News-Fair.

**Disambiguation** Even though ReFinED is the best disambiguation system in our evaluation, there is still room for improvement, particularly on metonym mentions, where it has an average error rate of 30.8%, but also on partial name and rare mentions. Given that ReFinED is among the best systems in these categories, we conclude that these categories are particularly hard to solve and are worth a closer look when designing new entity linking systems. Especially since they appear frequently in many benchmarks, as shown in Table 3.

**Reproducibility** We were able to reproduce the results reported on ReFinED's GitHub page for the AIDA-CoNLL test set and the updated MSNBC dataset with minor deviations of $\leq 0.6\%$. We achieved higher results than those reported in the paper on all evaluated benchmarks, since for the paper an older Wikipedia version was used (as noted by the authors on their GitHub page).

## 6.2 REL

Van Hulst et al. (2020) introduce REL (Radboud Entity Linker). REL uses Flair (Akbik et al., 2018) as default ER component which is based on contextualized word embeddings. For disambiguation, they combine local compatibility, (e.g., prior probability and context similarity), with coherence with other linking decisions in the document using a neural network that is trained on the AIDA-CoNLL training dataset. REL comes in two versions: one is based on a Wikipedia dump from 2014 and one is based on a dump from 2019. We evaluate the 2014 version because it outperforms the 2019 version on all our benchmarks except Wiki-Fair.

**Evaluation summary** REL achieves a high overall F1 score on all benchmarks and performs particularly well in the ER task. In the disambiguation task, it is outperformed by ReFinED and GENRE and performs poorly on Wiki-Fair. In the following we focus on weaknesses we found in the system.

**Recognition** REL has a high number of FPs for mentions where the ground truth entity is NIL. While on AIDA-CoNLL this is also due to outdated ground truth annotations, the trend is consistent across all benchmarks and indicates that REL could benefit from predicting NIL entities.

REL tends to detect mention spans that are shorter than those annotated in the ground truth;

---

[13]https://elevant.cs.uni-freiburg.de/emnlp2023/

[14]On News-Fair and Wiki-Fair (which we annotated with Wikidata entities), the 33M version is slightly better than the 6M version.

see the "partially included" column in Table 4.

**Disambiguation** REL performs well in the disambiguation task, except on Wiki-Fair, where it just barely outperforms our simple baseline. Many of the disambiguation errors fall into none of our specific error categories (Table 4), which is typically a hint that the true entity was not contained in the system's knowledge base and thus could not be predicted. This theory is supported by the fact that the REL version based on a Wikipedia dump from 2019 performs better on Wiki-Fair (and only on Wiki-Fair) than the 2014 version (Wiki-Fair is based on a Wikipedia dump from 2020).

REL also has trouble disambiguating partial names on Wiki-Fair, but it does not have that problem on the other benchmarks.

**Reproducibility** We were able to reproduce the results reported in the paper for most benchmarks within a margin of error of < 1.0%.

### 6.3 GENRE

GENRE (De Cao et al., 2021b) is an autoregressive language model that generates text with entity annotations. The generation algorithm is constrained so that the model generates the given input text with annotations from a fixed set of mentions and fixed candidate entities per mention. GENRE comes in two variants: A model that was trained on Wikipedia only and one that was fine-tuned on the AIDA-CoNLL dataset. We evaluate the fine-tuned version because it outperforms the Wikipedia version on all benchmarks in our evaluation.

**Evaluation summary** GENRE performs well on all benchmarks, but is typically outperformed by ReFinED and REL. GENRE has a relatively weak ER F1, but strong disambiguation accuracy. This indicates that it tends to annotate only those mentions for which it is confident that it knows the correct entity.

**Recognition** GENRE's ER F1, averaged over all benchmarks, is 8.5% worse than that of the best system (ReFinED). Precision is always better than recall, with an especially large difference on News-Fair and Wiki-Fair. Most other linkers show this discrepancy on those two benchmarks, but GENRE trades precision for recall more aggressively. Thanks to this, GENRE is among the systems with the lowest number of ER false positives and it is also very good at not linking mentions where the ground truth entity is NIL.

**Disambiguation** GENRE is the best system at disambiguating demonyms and is only beaten by REL at disambiguating metonyms. Both kinds of mentions appear often in the AIDA-CoNLL dataset it was fine-tuned on.

Even though GENRE disambiguates metonyms, partial names and rare mentions comparatively well, there is still room for improvement for these hard categories; see the respective comment in the discussion of ReFinED.

**Reproducibility** We could reproduce the result on the AIDA-CoNLL benchmark with a discrepancy of 0.7%. On the other benchmarks, the GENRE model trained on Wikipedia only is reported to give the best results, but performs very poorly in our evaluation; see this GitHub issue.

### 6.4 Ambiverse

Ambiverse uses KnowNER (Seyler et al., 2018) for ER and an enhanced version of AIDA (Hoffart et al., 2011) for entity disambiguation. KnowNER uses a conditional random field that is trained on various features such as a prior probability and a binary indicator that indicates whether the token is part of a sequence that occurs in a type gazetteer. The AIDA entity disambiguation component uses a graph-based method to combine prior probabilities of candidate entities, the similarity between the context of a mention and a candidate entity, and the coherence among candidate entities of all mentions.

**Evaluation summary** Ambiverse is outperformed by newer systems, even on its "own" benchmark AIDA-CoNLL (created by the makers of Ambiverse). On News-Fair and Wiki-Fair, its overall F1 score is hardly better than the baseline.

**Recognition** Ambiverse's ER component tends to recognize smaller spans than those from the ground truth[15]. However, the detected shorter spans are often still linked to the correct entity, as shown by a relatively high number of "wrong span" errors on News-Fair and Wiki-Fair.

Ambiverse has a high number of ER false positives for mentions where the ground truth entity is NIL across all benchmarks, which indicates that the system could benefit from predicting NIL entities.

**Disambiguation** Ambiverse performs relatively well on partial names on all benchmarks. This shows particularly on KORE50, where 61% of mentions are partial names. Apart from that, its

---

[15]For example, in the mention "1936 Summer Olympics", it detects only "Summer Olympics"

disambiguation is mediocre, with problems in the "demonym" and "metonym" category. This shows particularly on AIDA-CoNLL, where these two categories are most strongly represented.

**Reproducibility** Since Ambiverse uses a modified version of the systems introduced in Seyler et al. (2018) and Hoffart et al. (2011), no direct comparison to results reported in a paper is possible. However, the benchmark on which Ambiverse achieves its highest ranking in our evaluation is KORE50, which is a benchmark that was hand-crafted by the same research group that created Ambiverse. On the other hand it also has one of its lowest rankings on the AIDA-CoNLL test set which was also created by this research group.

### 6.5 Neural EL

This section has been moved to the appendix (A.3.1) due to limited space.

### 6.6 TagMe

This section has been moved to the appendix (A.3.2) due to limited space.

## 7 Conclusions

Our in-depth evaluation sheds light on the strengths and weaknesses of existing entity linking systems, as well as on problems with existing benchmarks (in particular, the widely used AIDA-CoNLL) and reproducibilty issues. We introduce two new benchmarks with clear annotation guidelines and a fair evaluation as primary goals.

In particular, we find that even the best systems still have problems with metonym, partial name and rare mentions. All linkers have troubles with non-named entities. They either ignore non-named entities completely or link too many of them. ReFinED performs best on almost all benchmarks including our independently designed and fair benchmarks. Several systems have reproducibility issues. The two newest systems, ReFinED and REL, are significantly better in that respect.

Our evaluation was more extensive than what we could fit into nine pages and we identified several frontiers for going deeper or further: describe more systems in detail, provide even more detailed numbers, include systems which only do disambiguation, evaluate also by entity type, and consider other knowledge bases; see Section 9.

## 8 Author Contributions

All three authors conducted the research. N.P. and M.H. annotated the benchmarks. M.H. implemented the evaluation of GENRE and Efficient EL, N.P. implemented the evaluation of the other linkers. N.P. is the lead developer of ELEVANT and implemented several extensions needed for the evaluation in this paper. All three authors wrote the paper, with N.P. taking the lead and doing the largest part.

## 9 Limitations

We only evaluated systems that perform end-to-end entity linking, which we consider the most relevant use case. However, more systems exist which do only entity recognition or only entity disambiguation, and these systems could be combined to perform entity linking.

We only evaluated systems with either code and trained models or an API available, and we could only evaluate the available versions. Our results often deviate from the results reported in the papers, sometimes significantly. For example, the GENRE model trained on Wikipedia is reported to give good results on many benchmarks, but the model provided online performs very poorly. The Efficient EL model was only trained on AIDA-CoNLL and could benefit from training on a larger and more diverse dataset (see Section A.3.4 for a detailed evaluation of Efficient EL). Re-implementing or re-training models from the literature is out of scope for this paper.

We only considered benchmarks and linkers with knowledge bases that are linkable to Wikidata, such as Wikipedia. However, in other research areas, there exist many knowledge bases and linkers for special use cases, e.g., biology or materials science. Outside of academia, the situation is even more complicated because the data is often proprietary (and sometimes also the employed software).

We would like to have reported results on more benchmarks, for example, Derczynski (Derczynski et al., 2015) and Reuters-128 (Röder et al., 2014), but had to restrict our analysis due to limited space. We selected the most widely used benchmarks.

The evaluation tool ELEVANT by Bast et al. (2022) allows to evaluate and compare the performance of entity linkers on a large selection of entity types (the usual ones: person, location, organization; but also many others). We limited our analysis

to the different error categories, which we found more (r)elevant.

We evaluate end-to-end entity linking results, which means that the disambiguation performance can only be evaluated on mentions that were correctly detected by a linker. Therefore, each linker's disambiguation performance is evaluated on a different set of ground truth mentions, thereby limiting the comparability of the resulting numbers. For example, a linker that detects only the mentions it can disambiguate well would achieve an unrealistically high disambiguation accuracy (at the cost of a low ER recall). A preferable way of evaluating the disambiguation performance would be to disentangle the ER and disambiguation components of each linker, and to evaluate the disambiguation component's accuracy on all ground truth mentions. However, this would require major changes to the linkers' code and might not be possible for all linkers.

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

# A Appendix

## A.1 Precision, recall, F1 score

We use precision, recall and F1 score to evaluate the entity linking systems. True positives (TP) are the linked mentions where the exact same text span is linked to the same entity in the ground truth. False positives (FP) are the linked mentions where either the span is not annotated in the ground truth or linked with a different entity. False negatives (FN) are ground truth mentions where either the span is not recognized by a system or linked with a wrong entity. A ground truth span that is recognized but linked with the wrong entity counts as both false positive and false negative. Optional entities count as neither true positive nor false negative. Unknown entities (i.e. entities that are linked to NIL) do not count as false negatives when they are not detected. Precision is defined as $\frac{TP}{TP+FP}$ and recall as $\frac{TP}{TP+FN}$. F1 score is the harmonic mean of precision and recall.

We also evaluate the ER capabilities of the systems. Here we only compare the predicted mention spans with the ground truth spans, regardless of the linked entities. Precision, recall and F1 score are defined as above.

## A.2 Excluded benchmarks

The following benchmark was excluded from our evaluation due to problems in the benchmark design:

• DBpedia Spotlight (Mendes et al., 2011): A small benchmark containing 35 paragraphs from New York Times articles from eight different categories. The annotators were asked to annotate "all phrases that would add information to the provided text". The result is a benchmark in which 75% of annotations are non-named entities. The benchmark contains annotations for words like "curved", "idea", or "house". On the other hand, phrases like "story", "Russian" or "Web language" are not annotated (even though "Web" and "Web pages" are) which makes the annotation decisions seem arbitrary.

## A.3 Evaluation of additional systems

### A.3.1 Neural EL

Gupta et al. (2017) introduce Neural EL, a neural entity linking system that learns a dense representation for each entity using multiple sources of information (entity description, entity context, entity types). They then compute the semantic similarity between a mention and the embedding of each entity candidate and combine this similarity with a prior probability to a final score. Neural EL focuses on entity disambiguation. The provided code is however also capable of performing end-to-end entity linking[16], which we are evaluating here.

---

[16]In the paper, Stanford-NER is used to evaluate the end-to-

**Evaluation summary** Neural EL achieves a low overall F1 score over all benchmarks. Its ER component performs decent on benchmark that contain only named entities, but weak on News-Fair and Wiki-Fair. Neural EL performs particularly weak in disambiguating partial names but solid in disambiguating demonyms.

**Recognition** Neural EL has a relatively high ER precision. Neural EL's ER system is particularly strict with linking only named entities which results in a high number of lowercased ER FNs and a generally low performance on all benchmarks containing lowercased entities.

**Disambiguation** Neural EL performs decent on demonyms. On our two benchmarks with a significant number of demonyms, Neural EL ranks 3rd and 4th in the demonym category.

On all benchmarks, Neural EL makes a high number or partial name errors. Only our baseline typically performs worse in this category.

**Reproducibility** We compare the results we achieved on the AIDA-CoNLL development and test set using the publicly available code with the results reported in Gupta et al. (2017). For the comparison, we provide ground truth mention spans to the system and exclude NIL links in both the ground truth and the predictions. However, we fall short of reproducing the reported results by 4.1% on the test set (78.8% vs. 82.9% reported in the paper) and by 7% on the development set (77.9% vs. 84.9% reported in the paper).

### A.3.2 TagMe

Ferragina and Scaiella (2010) propose TagMe, an entity linker designed to work well on very short texts like tweets or newsfeed items. They consider Wikipedia hyperlink anchor texts as possible mentions. For the disambiguation, they compute the relatedness between a mention's candidate entities and the candidate entities of all other mentions in the text and combine it with the prior probability of a candidate.

**Evaluation summary** TagMe frequently predicts non-named entities. Its overall F1 score is therefore low on benchmarks that contain only named entities. It achieves decent results in the overall disambiguation category which can partly be explained by the system ignoring mentions that are difficult to disambiguate. When filtering out non-named entity predictions, TagMe remains a weak end entity linking task.

system but beats our baseline on most benchmarks. TagMe leaves it up to the user to balance recall and precision with a configurable threshold.

**Recognition** TagMe has the lowest ER F1 scores on all benchmarks with particularly low precision. Recall is low on benchmarks containing only named entities, but decent on News-Fair and Wiki-Fair.

TagMe's ER component has a tendency towards including more tokens in its detected spans than what is annotated in the ground truth, thus achieving good results in the "partially included" category. On AIDA-CoNLL and MSNBC where this effect is most observable, this can however often be ascribed to erroneous benchmark annotations[17].

TagMe produces a relatively high number of ER FP errors in the "wrong span" category, although sometimes these errors could also be attributed to debatable ground truth spans or missing alternative ground truth spans in the benchmark[18].

**Disambiguation** TagMe performs decent in the overall disambiguation category and shows a weak disambiguation performance only on AIDA-CoNLL. The weak performance on AIDA-CoNLL can be attributed to a high number of metonym errors on this benchmark as well as a generally high number of demonym errors. A closer inspection shows that TagMe has a tendency to falsely link demonyms to the corresponding language[19].

TagMe has a relatively low number of disambiguation errors in the "partial name" category on most benchmarks, especially on KORE50. Since partial names make up 61% of mentions on KORE50, this results in TagMe being the second-best performing system on KORE50 in the overall disambiguation category. However, it also has the lowest ER recall on KORE50. Comparing the individual predictions to those of Ambiverse shows that 24 out of 28 partial name mentions that Ambiverse disambiguates wrongly are either not detected by TagMe or also disambiguated wrongly.

**Reproducibility** We evaluated TagMe over the WIKI-ANNOT30 dataset used in the original paper to evaluate end-to-end linking. Since we were unable to reconstruct the original train and test splits

---

[17]E.g., TagMe links the entire phrase in "2003 Kids' Choice Awards" to the entity "2003 Kid's Choice Awards" which is not actually an error.

[18]E.g., TagMe links the entire phrase in "London Heathrow airport".

[19]E.g., in "Polish citizens" linking "Polish" to the Polish language rather than the country.

of the dataset, we used the entire dataset for evaluation. However, we fall short of reproducing the F1 score reported in the original TagMe paper by almost 20% using the official TagMe API (57.5% vs. 76.2% reported in the paper).

### A.3.3 DBpedia Spotlight

Mendes et al. (2011) propose DBpedia Spotlight, an entity linking system that aims specifically at being able to link DBPedia entities of any type. DBpedia identifies mentions by searching for occurrences of entity aliases. Candidate entities are determined based on the same alias sets. For the disambiguation, DBpedia entity occurrences are modeled in a Vector Space Model with TF*ICF weights where TF is the term frequency and represents the relevance of a word for a given entity and ICF is the inverse candidate frequency which models the discriminative power of a given word. Candidate entities are ranked according to the cosine similarity between their context vectors and the context of the mention. An improved version of the system was introduced in (Daiber et al., 2013).

**Evaluation summary** DBpedia Spotlight is an entity linking system dedicated to linking entities of all types including non-named entities. When adding it to our set of evaluated linkers, it is the weakest performing system on almost all benchmarks including those containing non-named entities. This can mostly be attributed to the weak performance of the ER component, but its disambiguation results are not convincing either. DBpedia Spotlight comes with multiple configurable parameters such as a confidence threshold to balance precision and recall and thus, similar to TagMe, leaves it to the user to find a good parameter setting.

**Recognition** DBpedia Spotlight has the lowest ER precision on almost every benchmark, mainly because it falsely detects too many lowercased mentions. While some ER FPs stem from DBpedia Spotlight trying to solve a different task than what most benchmarks were designed for[20], other errors are clearly not what is desired under any task description[21]. When filtering out lowercase predictions, ER precision improves, but is still among the lowest on all benchmarks.

DBpedia Spotlight achieves the highest ER recall on News-Fair and the second-highest on Wiki-Fair (only outperformed by ReFinED) due to the low number of undetected lowercase mentions. On all other benchmarks, ER recall is mediocre.

DBpedia Spotlight makes the most ER FNs in the "partially included" category[22] on all benchmarks except KORE50 (REL performs worse).

**Disambiguation** DBpedia Spotlight performs particularly weak at disambiguating partial names and rare entities. The latter typically indicates that a system relies heavily on prior probabilities and does not put enough emphasis on the context of the mention[23].

**Reproducibility** We tried to reproduce the results reported in the original paper on the DBpedia Spotlight benchmark using the official DBpedia Spotlight API. We were unable to reproduce the results for no configuration which we interpreted as using default parameters (42.4% vs. 45.2% reported in the paper). We were also unable to reproduce the results reported for the best configuration, which we assume corresponds to a confidence threshold of 0.35 and a support of 100 as indicated in the paper (33.6% vs. 56% reported in the paper). However, it is important to note, that the system has undergone many changes since its first publication.

### A.3.4 Efficient EL

Efficient EL (De Cao et al., 2021a) is a generative model with parallelized decoding and an extra discriminative component in the objective. The provided model is only trained on the AIDA-CoNLL training data, and the paper evaluates only on the AIDA-CoNLL test set.

**Evaluation summary** When adding it to our set of evaluated linkers, Efficient EL is only outperformed by ReFinED on AIDA-CoNLL but performs very poorly on all other benchmarks, since it was only trained on AIDA-CoNLL. We therefore only evaluate its performance on AIDA-CoNLL. On this benchmark, it has the best ER system, but GENRE is better on some disambiguation categories, leaving room for improvement of Efficient EL.

**Recognition** Efficient EL is very good at detecting long mentions and has the lowest number of ER

---

[20]E.g., in "sports events" linking "sports" to the entity for "sport".
[21]E.g., in "Spanish police" linking "Spanish police" to the entity for Spain.

[22]E.g., in "South Korea" linking both "South" and "Korea" to the corresponding country.
[23]E.g., in the phrase "in Columbus, Ohio", Columbus is linked to the entity of Christopher Columbus instead of Columbus the capital city of Ohio.

FPs on AIDA-CoNLL.

**Disambiguation** Efficient EL's disambiguation accuracy on AIDA-CoNLL is close to that of GENRE and REL but it is significantly outperformed by ReFinED in that category.

Efficient EL is the best demonym and rare entity disambiguator on AIDA-CoNLL, but is significantly worse at disambiguating metonyms and partial names then ReFinED, GENRE and REL.

**Reproducibility** The paper only reports results on the AIDA-CoNLL test set. The result in our evaluation is close, but not equal to the result in the paper (85.0% F1 score compared to 85.5% in the paper).

### A.4 Annotation guidelines

**What to annotate:** Only annotate entities that are an instance of at least one of our whitelist types or an instance of a subclass of one of the whitelist types.

**Quantities and datetimes:** Annotate quantities (including ordinals) and datetimes with a special label QUANTITY or DATETIME. Units should not be included in the mention.

**Demonyms:** In general, annotate demonym mentions with the country. Additionally, annotate the mention with the ethnicity or country-citizens if the culture or ethnicity is being referred to (e.g., "[American] dish"). The mention should not be annotated with the ethnicity in cases like "[Soviet]-backed United Arab Republic" (Soviet refers to (a part of) the government which is better represented by the country) or "[American] movie" (it's still an American movie if the director decides to migrate to another country). Only annotate the mention with the language if it is obvious that the language is being referred to (e.g., '"sectores" means "sectors" in [Spanish]').

**Spans:** Use the Wikipedia title as mention. If in doubt, also allow other spans that are aliases for the referenced entity. If an argument could be made for splitting a mention into several, annotate the splitted version as an alternative (e.g., "[[Louis VIII], [Landgrave of Hesse-Darmstadt]]").

**Optional mentions:** Use optional mentions for cases where the entity name and not the entity itself is being referred to, e.g., "known generally as the [stirrup dart moth]".

**NIL entities:** Annotate entities not in Wikidata with *Unknown* to evaluate ground truth NIL errors and support coreference resolution evaluation for entities linked to NIL.

**Coreferences:** A coreference is when the name of an entity that appears elsewhere in the document is not repeated but replaced by a pronoun/description for solely linguistic purposes. E.g., "Barack Obama's wife" should not be annotated unless Michelle Obama is explicitly mentioned elsewhere in the document, because only then it's a coreference. Otherwise it's a second-order entity linking problem and we're not evaluating that.

### A.5 Type Whitelist

To ensure a consistent annotation of entities in our benchmark, we annotated all entities that are an instance or an instance of a subclass of one of the types in a type whitelist. In rare cases where the Wikidata class hierarchy was clearly erroneous, we deviated from this annotation policy. The following is a complete list of these whitelist types with their Wikidata QID:

Person (Q215627), Fictional Character (Q95074), Geographic Entity (Q27096213), Fictional Location (Q3895768), Organization (Q43229), Creative Work (Q17537576), Product (Q2424752), Event (Q1656682), Brand (Q431289), Genre (Q483394), Languoid (Q17376908), Chemical Entity (Q43460564), Taxon (Q16521), Religion (Q9174), Ideology (Q7257), Position (Q4164871), Occupation (Q12737077), Academic Discipline (Q11862829), Narrative Entity (Q21070598), Award (Q618779), Disease (Q12136), Religious Identity (Q4392985), Record Chart (Q373899), Government Program (Q22222786), Human Population (Q33829), Color (Q1075), Treatment (Q179661), Symptom (Q169872), Anatomical Structure (Q4936952), Sport (Q349), Animal (Q729).