# OpenReview forum: "A Fair and In-Depth Evaluation of Existing End-to-End Entity Linking Systems"
_EMNLP/2023/Conference — EMNLP 2023 Main_

### Official Review · Reviewer_wuuS · 2023-07-20

**Typos Grammar Style And Presentation Improvements:** 1. It would be beneficial to place th…
**Soundness:** 4

**Excitement:**

4: Strong: This paper deepens the understanding of some phenomenon or lowers the barriers to an existing research direction.

**Paper Topic And Main Contributions:**

This paper argues that existing evaluations have limited practical significance in assessing the practical performance of entity linking (EL) systems. as they only provide coarse evaluation metrics and suffer from artifacts and biases. The authors provide an evaluation with two benchmarks and in-depth analysis of existing EL systems and benchmarks.

**Questions For The Authors:**

Q. A. 362-369 implies that there are differences in the metric calculation among different systems. Perhaps it is necessary to provide another evaluation that completely disregards date and quantity for fairer comparison?

Q. B. Table 4 only provides average results over all benchmarks. I think it would be beneficial to include metrics on separate benchmarks, allowing readers to make comparisons across various benchmarks, especially when considering newly introduced ones. Additionally, given the variations in the design and scale of these benchmarks, the appropriateness of simply averaging all the results may needs further discussion.

**Reasons To Accept:**

1. The author discussed a widespread phenomenon in current entity linking evaluations, which was overlooked and inadequately discussed.
2. Rich and in-depth analysis.
3. New benchmarks with annotation alternatives for fair comparsions.

**Reasons To Reject:**

1. Lacking evaluation results on separate benchmarks.  (Please refer to question B)
2. Some parts of the text might be a little less reader friendly. (Please refer presentation improvements)

**Reproducibility:**

4: Could mostly reproduce the results, but there may be some variation because of sample variance or minor variations in their interpretation of the protocol or method.

**Reviewer Confidence:**

4: Quite sure. I tried to check the important points carefully. It's unlikely, though conceivable, that I missed something that should affect my ratings.

---

> ### Author Rebuttal · Authors · 2023-08-29
>
> Thank you very much for your review. Here is our brief response to each of your questions and criticisms.
>
> **@Question A: "[...] provide another evaluation that completely disregards date and quantity for fairer comparison?”:** In a sense, that is exactly what we are doing. If one just *omits* the ground truth labels for dates and quantities, a system that regards those as entities would be unfairly punished with false positives. Only by annotating such mentions as optional in our benchmarks, is it possible to not punish such false positives. Similarly, a system that does not regard dates and quantities as entities is not punished with false negatives. Including dates and quantities as optional ground truth annotations is therefore arguably fairer.
>
> **@Question B: ”Lacking evaluation results on separate benchmarks.”:** We absolutely agree that the reader should have access to the non-averaged results. However, space is limited and a table with results for all error categories over all benchmarks for all linkers would be huge. We therefore include these results via a link to an interactive online evaluation tool as noted in line 394-395 and footnote 10 (we only include the link in the final version of the paper for anonymity reasons). This facilitates the inspection of the results for the reader as opposed to including a single huge static table in the paper.
>
> **@"Metrics section closer to Table 4":** This conflicts with two other requirements: (1) Table 4 should be close to section 6, since this section revolves around table 4. (2) The Metrics section should precede the Benchmarks section, since we refer to some metrics in the Benchmarks section.
>
> **@”readability of Figure 1”:** Thanks for pointing this out, we will fix that in the final version of the paper.
>
> **@”present baseline and other systems separately”:**  We thought about that, however, we find it more useful to have the systems sorted by overall F1 score. This way it is immediately clear that some systems are even outperformed by a simple baseline.
>
> **@Missing \item in line 128:** The first sentence is a concise summary of the contributions. We tried to make that clear by adding "More specifically:" in line 132, followed by the two bullet points describing our main contributions.

---

### Official Review · Reviewer_JFFU · 2023-08-02

**Soundness:** 4

**Excitement:**

4: Strong: This paper deepens the understanding of some phenomenon or lowers the barriers to an existing research direction.

**Missing References:**

There is one missing reference that I think absolutely needs to be added, and another one that's optional (but I do think it should also be considered - even though it is in the context of slot filling rather than NER/NEL):

Ortmann, K. (2022, June). Fine-Grained Error Analysis and Fair Evaluation of Labeled Spans. In Proceedings of the Thirteenth Language Resources and Evaluation Conference (pp. 1400-1407).

Vashishth, S., Joshi, R., Prayaga, S. S., Bhattacharyya, C., & Talukdar, P. (2018). Reside: Improving distantly-supervised neural relation extraction using side information. arXiv preprint arXiv:1812.04361.

**Paper Topic And Main Contributions:**

The paper is generally well-written and illustrated. Sufficient details are provided to reproduce most of the work even if implementing from scratch.

One general observation is that the paper does not seem to adhere to the common terminology in the field: e.g., NERL and NEL are the well-known name of the tasks instead of ER and EL. Using other task names than the common ones turns the whole experience of reading the paper into a struggle.

This is one of the first papers that actually proposes the idea of fair evaluation for NEL, therefore its omission is not great.

There are several comments that I think need to be addressed:
1) There is no need for multiple sections in the introduction
2) There is no need for so many subsections for all the benchmarks in Section 4 - a compact (or dense) writing style is better suited for a scientific publication.
3) Title of section 3 is problematic as it suggests that the authors were the first to propose this, even though many proposals have been encountered during the years. In particular when treating NER and NEL one should not omit CSSF or SF (Cold Start Slot Filling or simply Slot Filling). If taking the larger field into account there have been many attempts to propose fair evaluations, starting with the TAC-KBP scorers from the TAC challenges.
4) Not immediately clear what are the final F1-scores of the fair evaluations - or if there is anything equivalent to an improved F1-metric.
5) Section 6 only discusses 4 systems instead of creating a table in which to synthetize the results from all the systems. Moving some systems in the Appendix creates a problem - as it might look like these other systems were dismissed.
6) The limitations section mentioned that the analysis was limited to certain error categories which the authors found relevant without mentioning the criteria they used for relevance.

I have read the rebuttal and think they have done a good job of explaining their thinking process. However, I will not modify my review scores.

**Questions For The Authors:**

Q1: Please clarify what is the metric that should replace the F1 metric?
Q2: Please explain why you have focused only on NER/NEL when framing this issue, given the fact that similar problems and solutions were also proposed for relation extraction and slot filling?

**Reasons To Accept:**

- a new attempt to create fair evaluations for NER/NEL
- new metrics for NER/NEL that include false negatives and false positiggbes, as well as disambiguation error rate
- overall good readability

**Reasons To Reject:**

- NER/NEL should not be discussed without the general context of relation extraction / slot filling today
- some missing bibliography
- not clear what is the overall metric that should be used from now on in fair evaluations

**Reproducibility:**

4: Could mostly reproduce the results, but there may be some variation because of sample variance or minor variations in their interpretation of the protocol or method.

**Reviewer Confidence:**

5: Positive that my evaluation is correct. I read the paper very carefully and I am very familiar with related work.

**Typos Grammar Style And Presentation Improvements:**

Generally they should adhere to the terminology and abbreviations of modern NLP: NER (instead of ER), NEL (instead of EL), etc.

---

> ### Author Rebuttal · Authors · 2023-08-29
>
> Thank you very much for your review. Here is our brief response to each of your questions and criticisms.
>
> **@"Q1: Please clarify what is the metric that should replace the F1 metric?"  and "it is not clear what the final F1 scores are":** We are not sure that we fully understand this comment. The main point of our paper is not that the F1 metric should be replaced by another single-valued metric. The point of our paper is to take a much more refined look at the problem, which we do. Having said that, our benchmarks allow for alternative annotations and optional annotations, which indeed requires changes to the F1 score. Footnote 8 explains how we deal with alternative annotations. Lines 362-369 explain how we deal with optional annotations. The F1 score is then computed with the usual formulas mentioned in Appendix A.1.
>
> **@"Q2: Please explain why you have focused only on NER/NEL" and "one should not omit CSSF or SF (Cold Start Slot Filling or simply Slot Filling)":** Entity linking is a fundamental problem with literally thousands of papers about it. The paper is already packed with content as it is, and we consider it quite reasonable to not make the scope of the paper even wider. Problems like CSSF or SF or relation extraction are distinctly different problems in the context of knowledge bases. Of course, they are related to entity linking, just like many other NLP problems are related to entity linking because entity linking is so fundamental.
>
> **@"the paper does not seem to adhere to the common terminology in the field [...] NERL and NEL are the well-known names of the tasks":** We chose the name ER and EL fully deliberately because the N stands for "named" and considering non-named entities is an important part of our study. Also, EL is very common in the literature (2200 results in Google Scholar for EL "entity linking"), more common even than NEL (620 results for NEL “named entity linking”) and much more common than NERL (only 29 results for NERL "named entity recognition and linking").
>
> **@"This is one of the first papers that actually proposes the idea of fair evaluation for NEL, therefore its omission is not great.":** We don't understand which "omission" you are talking about here. If you mean the terminology, our choice was deliberate and we don't think that criticism is justified, see the comment above.
>
> **@"no need for multiple sections in the introduction", "no need for so many subsections in Section 4":** We deliberately introduced two subsections in the introduction to give it some structure due to its length of 1.5 pages. Section 4 has one subsection per benchmark in order to follow the same format as Section 6, which has one subsection per evaluated system.
>
> **@"title of section 3 is problematic":** We don't claim that this is novel, this is just a needed preliminary for the following sections. See our "Contributions" section for what we consider as our contributions. In particular, Section 3 is not mentioned there.
>
> **@"Section 6 only discusses 4 systems":** We would have loved to add more, but the space was limited and the paper is already packed with information, given the topic and the breadth of our goal. We therefore discuss the four best systems (see Table 1) in the main text and the others in the appendix.
>
> **@"the analysis was limited to certain error categories":** This is a misunderstanding, please read the corresponding sentence again. It doesn’t say that we omitted any error categories. It says that we preferred a breakdown of our analysis by error categories to a breakdown by entity types, for the simple reason that the former is much deeper and more insightful than the latter. We would have loved to include the latter as well, but this was not possible in the limited space.
>
> **@missing references:** Thank you for the (Ortmann, 2022) reference. It is not central, but indeed relevant for a detailed aspect of the ER part of the problem. The reference (Vashishth, 2018) is about a particular method for relation extraction, which is only remotely related to our paper (see our comment on CSSF and SF above).

---

### Official Review · Reviewer_PBJb · 2023-08-07

**Soundness:** 4

**Excitement:**

3: Ambivalent: It has merits (e.g., it reports state-of-the-art results, the idea is nice), but there are key weaknesses (e.g., it describes incremental work), and it can significantly benefit from another round of revision. However, I won't object to accepting it if my co-reviewers champion it.

**Paper Topic And Main Contributions:**

In response to the problems existing in the evaluation systems of the end-to-end entity linking task, this paper proposes a new evaluation system. For evaluation methods using only general aggregate measures, this paper proposes a detailed error analysis, including: lowercased, partially included, ground truth NIL, wrong span and other indicators. Then, in view of the limitations of the current mainstream benchmarks, two new benchmarks are proposed: Wiki-Fair and News-Fair. Finally, the current entity link system and model are re-evaluated on the new benchmarks. Although the size and variety of newly built benchmarks are limited, they still bring new contributions to the entity link community.

**Questions For The Authors:**

Question A: How is the problem of imbalanced entity types addressed when constructing new benchmarks? The problem of new benchmarks seems to still exist.
Question B: Why is it not sampled from other sources to enrich the entity types of benchmarks?
Question C: When building new benchmarks, how to sample each category of entity?
Question D: When building new benchmarks, do you consider overlap entities?


**Reasons To Accept:**

- This paper systematically analyzes the limitations of the current mainstream benchmarks in the field of entity links, which are crucial for entity linking research. This paper provides a systematic analysis of the research in this field.
- In view of the limitations of existing benchmarks, this paper proposes two new benchmarks to alleviate these problems and reveal several artifacts and biases.
- this paper is written structured and easy to follow.


**Reasons To Reject:**

- The benchmarks proposed in this paper seem to be relatively small, and the entity categories covered are not rich enough.
- There seems to be no good response to some of the benchmark limitations mentioned above, such as the imbalance of entity types in benchmark.
- The writing skills of some details need to be improved.


**Reproducibility:**

5: Could easily reproduce the results.

**Reviewer Confidence:**

4: Quite sure. I tried to check the important points carefully. It's unlikely, though conceivable, that I missed something that should affect my ratings.

**Typos Grammar Style And Presentation Improvements:**

-	The image quality of Figure 1 seems to be not clear enough. It is recommended to use a vector graphic.
-	More details need to be listed in Table 2, such as the distribution of word counts for mentions, distribution of types, and so on.
-	Section 6 seems to have too much introduction about the models. The paper should focus on introducing the analysis results of the benchmarks.

---

> ### Author Rebuttal · Authors · 2023-08-29
>
> Thank you very much for your review. Here is our brief response to each of your questions and criticisms.
>
> **@"Question A: How is the problem of imbalanced entity types addressed when constructing new benchmarks?":** We chose the articles at random and just three random paragraphs from each article. Our goal was not to have balanced entity types, but to avoid strong imbalances, as in the existing benchmarks. For example, as you can see from the last three columns of Table 3, in the existing benchmarks almost all entities are of the types person, location, organization. These are indeed common in text, but in our benchmarks, we also have 20 - 40% entities of other types.
>
> **@"Question B: Why not consider other sources to enrich the entity types":** We already cover random extracts from random Wikipedia articles and from random news articles. That already gives good coverage and was a lot of annotation work. When considering other sources, one quickly gets into application-specific issues (for example, Tweets or entity linking in biomedical texts). That is certainly worthwhile, too, but a whole new paper and out of scope for this one. Also, it was already very challenging to pack the contents we have now into the available space.
>
> **@"Question C: When building new benchmarks, how to sample each category of entity?":** See the answer to Question A.
>
> **@"Question D: When building new benchmarks, do you consider overlap entities?":** We do consider alternatives both for the recognition part (ER) and the disambiguation part (ED). But we only do that when the alternatives are more or less equivalent. When one alternative fits clearly better, we pick only the better one. For example, in the mention "2021 Summer Olympics", it is clearly better to link the whole mention to that specific event than to link "Olympics" to the generic "Olympic Games".
>
> **@"benchmarks proposed seem to be relatively small":** Our two new benchmarks are not large, but comparable in size to most existing entity-linking benchmarks, with the sole exception of AIDA-CoNNL, which is significantly larger. See Table 3 of our submission, column “mentions”.
>
> **@"the entity categories covered are not rich enough":** Telling from the rest of your review, we take it that you mean the entity types we consider. Our entity types cover *all* annotations in the benchmarks we consider (which are the most used ones in existing work on the entity-linking problem) as well as all entities in our new benchmarks that are not too abstract. If you miss an entity type, then the reason is very likely that there is no entity of that type in any of the benchmarks we considered (old and new).
>
> **@"The writing skills need to be improved":** We are experienced writers and have written the paper with great care. Without concrete suggestions, we don't know what to make of such criticism.

---

### Meta-Review · Area_Chair_37oA · 2023-09-16

**Recommendation:** 5

**Metareview:**

**Summary:**
The authors argue that current evaluations of entity linking (EL) systems lack practical significance due to their reliance on coarse metrics and vulnerability to artifacts and biases. This underscores the necessity for more comprehensive evaluation methods. Accordingly, the paper presents an innovative evaluation system that includes a detailed error analysis, encompassing various indicators. Furthermore, the paper highlights the limitations of traditional benchmarks by introducing two new ones, Wiki-Fair and News-Fair. These newly introduced benchmarks have been utilized to  re-evaluate existing entity linking systems and models.

**Strengths:**
The reviewers agree on the paper's strengths, which are as follows:
The author address a widespread phenomenon in current entity linking evaluations that has been previously overlooked and insufficiently discussed. This paper systematically analyzes the limitations of mainstream benchmarks, offering a comprehensive examination of research in this field.
The paper presents a novel attempt to create fair evaluations for NER/NEL, introducing new benchmarks and metrics that encompass false negatives and false positives, and disambiguation error rates.

**Weaknesses:**
The paper deviates from common terminology in the field, as NERL and NEL are recognized task names, rather than ER and EL. Additionally, discussing NER/NEL without considering the broader context of relation extraction and slot filling is unusual in current discourse. The proposed benchmarks appear relatively small, with limited coverage of entity categories. Some concerns regarding benchmark limitations, such as entity type imbalance, are not adequately addressed. Consequently, it remains unclear which overall metric should be adopted in fair evaluations, and there is a lack of evaluation results on separate benchmarks. Lastly, certain portions of the text is less reader-friendly.

**Author-Reviewer discussion and acknowledgment:**
Reviewers provided multiple comments and questions, all of which the authors addressed in their rebuttal response. The authors effectively explain their thought process. All the reviewers acknowledged the authors' rebuttal.

**Conclusion:**
The paper is generally well-written and well-illustrated. It provides sufficient details to reproduce most of the work. However, reviewers recommend that the authors include missing references in the bibliography, and some minor improvements in the writing of certain details would enhance the overall quality of the paper.

---

### Decision · Program_Chairs · 2023-10-07

**Decision:**

Accept-Main

**Comment:**

**Summary:**
The authors argue that current evaluations of entity linking (EL) systems lack practical significance due to their reliance on coarse metrics and vulnerability to artifacts and biases. This underscores the necessity for more comprehensive evaluation methods. Accordingly, the paper presents an innovative evaluation system that includes a detailed error analysis, encompassing various indicators. Furthermore, the paper highlights the limitations of traditional benchmarks by introducing two new ones, Wiki-Fair and News-Fair. These newly introduced benchmarks have been utilized to  re-evaluate existing entity linking systems and models.

**Strengths:**
The reviewers agree on the paper's strengths, which are as follows:
The author address a widespread phenomenon in current entity linking evaluations that has been previously overlooked and insufficiently discussed. This paper systematically analyzes the limitations of mainstream benchmarks, offering a comprehensive examination of research in this field.
The paper presents a novel attempt to create fair evaluations for NER/NEL, introducing new benchmarks and metrics that encompass false negatives and false positives, and disambiguation error rates.

**Weaknesses:**
The paper deviates from common terminology in the field, as NERL and NEL are recognized task names, rather than ER and EL. Additionally, discussing NER/NEL without considering the broader context of relation extraction and slot filling is unusual in current discourse. The proposed benchmarks appear relatively small, with limited coverage of entity categories. Some concerns regarding benchmark limitations, such as entity type imbalance, are not adequately addressed. Consequently, it remains unclear which overall metric should be adopted in fair evaluations, and there is a lack of evaluation results on separate benchmarks. Lastly, certain portions of the text is less reader-friendly.

**Author-Reviewer discussion and acknowledgment:**
Reviewers provided multiple comments and questions, all of which the authors addressed in their rebuttal response. The authors effectively explain their thought process. All the reviewers acknowledged the authors' rebuttal.

**Conclusion:**
The paper is generally well-written and well-illustrated. It provides sufficient details to reproduce most of the work. However, reviewers recommend that the authors include missing references in the bibliography, and some minor improvements in the writing of certain details would enhance the overall quality of the paper.